

# Efficient generation of human primordial germ cell-like cells from pluripotent stem cells in a methylcellulose-based 3D system at large scale

Xiaoman Wang[1,2,3,4,5,*], Tingting Liao[6,*], Cong Wan[2], Xiaoyu Yang[7], Jiexiang Zhao[2], Rui Fu[1], Zhaokai Yao[2], Yaping Huang[2], Yujia Shi[2], Gang Chang[8], Yi Zheng[2], Fang Luo[2], Zhaoting Liu[2], Yu Wang[1], Xinliang Mao[9] and Xiao-Yang Zhao[2,4,5]

[1] State Key Laboratory of Stem Cell and Reproductive Biology, Institute of Zoology, Chinese Academy of Sciences, Beijing, China
[2] Department of Developmental Biology, School of Basic Medical Sciences, Southern Medical University, Guangzhou, Guangdong, China
[3] College of Life Sciences, University of the Chinese Academy of Sciences, Beijing, China
[4] Guangdong Provincial Key Laboratory of Construction and Detection in Tissue Engineering, Southern Medical University, Guangdong, China
[5] Guangzhou Regenerative Medicine and Health Guangdong Laboratory (GRMH-GDL), Guangzhou, Guangdong, China
[6] Reproductive Medicine Center, Xiangya hospital, Central South University, Changsha, Hunan, China
[7] State Key Laboratory of Reproductive Medicine, Clinical Center of Reproductive Medicine, The First Affiliated Hospital of Nanjing Medical University, Nanjing, China
[8] Department of Biochemistry and Molecular Biology, Shenzhen University Health Science Center, Shenzhen, Guangdong, China
[9] School of Food Science and Engineering, South China University of Technology, Guangzhou, Guangdong, China
* These authors contributed equally to this work.

Corresponding authors
Yu Wang, wangyu@ioz.ac.cn
Xinliang Mao, xlmao918@foxmail.com

## ABSTRACT

**Background:** The mechanisms underlying human germ cell development and infertility remain largely unknown due to bioethical issues and the shortage of experimental materials. Therefore, an effective in vitro induction system of human primordial germ-like cells (hPGCLCs) from human pluripotent stem cells (hPSC) is in high demand. The current strategies used for the generation of hPGCLCs are not only costly but also difficult to perform at a large scale, thereby posing barriers to further research. In this study, we attempted to solve these problems by providing a new 3D culture system for hPGCLC differentiation.

**Methods:** The efficiency and relative yield of a methylcellulose (MC)-based 3D hPGCLC induction system were first compared with that of a conventional U96 system. Then, we examined the gene expression of germ cell marker genes and the key epigenetic modifications of the EpCAM-/INTEGRINα6-high cells from the 3D MC induction system and the U96 system via quantitative PCR and immunofluorescence. Finally, the reliability of the MC-based 3D hPGCLC induction system was evaluated via the generation of induced pluripotent stem cells (iPSCs) from the testicular cells of one patient with obstructive azoospermia (OA) and followed by the subsequent differentiation of iPSCs into the germ cell lineage.

**Results:** In the present study, we demonstrated that the 3D MC induction system combined with low-cell attachment plates facilitated the generation of hPGCLCs at a large scale. We found that the hPGCLCs generated via the MC system shared similar characteristics to that via the U96 system in terms of the gene expression profiles, germ cell-specific markers, epigenetic modification states and cellular states. In addition, hPGCLCs from iPSCs derived from one OA patient were generated with high efficiency via the present 3D MC induction system.

**Discussion:** The in vitro induction of hPGCLCs from human embryonic stem cells (hESCs)/human induced pluripotent stem cells (hiPSCs) has significant implications in exploring the underlying mechanisms of the origin and specification of hPGCs and the epigenetic programming of the human germ line as well as treating male infertility. Here, we developed a simple and efficient 3D induction system to generate hPGCLCs from hESCs/hiPSCs at a large scale, which facilitated the study of human germ cell development and stem cell-based reproductive medicine.

# INTRODUCTION

In mammals, including humans, the earliest germ cells are primordial germ cells (PGCs), which are derived from early embryos. After a precise and complicated developmental process, PGCs finally differentiate into spermatozoa or oocytes. Germ cells are crucial for the transmission of genetic and epigenetic information to the next generation. Therefore, defects in the germline may lead to infertility and other severe diseases (*Reik & Surani, 2015*). Although 10–15% of couples are affected by infertility (*Thonneau et al., 1991*), the mechanisms underlying human germ cell development and infertility remain unknown mainly because of bioethical issues and the shortage of experimental materials, such as human embryos. Accordingly, an effective in vitro human germ cell induction system or disease model is in high demand and will not only facilitate an understanding of human germline development but also provide a platform for treating disorders arising from human germ cell defects (*Nagamatsu & Hayashi, 2017*).

Recently, based on the knowledge of the in vivo mouse primordial germ cell (mPGC) specification, the in vitro mouse germ cell specification process was reconstituted from pluripotent stem cells (PSCs). The mouse PSCs were first induced into pre-gastrulation epiblast-like cells and then immediately differentiated into PGC-like cells (PGCLCs), which possessed the developmental capability for both spermatogenesis and oogenesis (*Hayashi et al., 2011*, *2012*). As germ cell induction system can provide abundant experimental materials, it has been utilized as a powerful platform to investigate the regulatory networks and dynamic epigenetic modifications during mPGC specification (*Aramaki et al., 2013*; *Kurimoto et al., 2015*; *Murakami et al., 2016*; *Nakaki et al., 2013*). Furthermore, based on this induction system, sperm-like cells or mature oocytes were generated in vitro through the PGCLCs derived from mouse ESCs/induced pluripotent
stem cells (iPSCs) (*Hikabe et al., 2016*; *Zhou et al., 2016*). And human PGCLCs human primordial germ-like cells (hPGCLCs) could also be induced from human PSCs (*Irie et al., 2015*; *Sasaki et al., 2015*). By employing hPGCLC induction systems, SOX17 and EOMES were identified to be critical factors for hPGCLC specification (*Chen et al., 2017*; *Irie et al., 2015*; *Kojima et al., 2017*), indicating that hPGCLC induction systems had great advantages in exploring the mechanisms underlying human germ cell development.

Although the in vitro hPGCLC induction systems recapitulated the development of hPGCs, some limitations still existed. During the induction of hPGCLCs, ultra-low cell attachment U-bottom 96-well plates or other similar plates were needed to enable the formation of embryoid bodies (EBs), followed by several days of suspension in induction medium. These steps are labor-intensive and dramatically limit the scale of hPGCLC production. On the other hand, many experiments are restricted for the insufficient production of hPGCLCs, including the proteome analysis of the dynamics of translation or the post-translation modification, which requires a large number of hPGCLCs. Therefore, an alternative solution was to employ a 3D culture system that allows more seeding cells to form embryonic bodies and thus saves both labor and medium. Collagen, methylcellulose (MC), and polymeric hydrogel biomaterials in combination with larger low-cell attachment plates were considered as the potential approaches to generate EBs efficiently (*Jiang et al., 2015*; *Kothapalli & Kamm, 2013*; *Otsuji et al., 2014*). However, whether they can be used to induce PGCLCs remains unclear.

Here, we reported a modified system to robustly generate hPGCLCs from human pluripotent stem cells (hPSCs) at large scale, and the yielded hPGCLCs exhibited typical gene expression and epigenetic modification profiles similar to that of human PGCs. Our work allows for the establishment of a new large-scale hPGCLCs induction strategy and has implications for deciphering the mechanisms underlying human germline development.

## MATERIALS AND METHODS

### Culture of hPSCs

The human embryonic stem cell (hESC)/hiPSC lines (Fy-hES-3, Fy-hES-12, YiPS-1, and YiPS-2) were cultured in feeder-free medium (CA1001500; PSCeasy®, CELLAPY, Beijing, China) on Matrigel (354277; Corning Inc., Corning, NY, USA). Cell media were changed daily. Cells were passaged every 5–7 days using ethylenediaminetetraacetic acid (EDTA) (CA3001500; PSCeasy®, CELLAPY, Beijing, China).

### Induction of iMeLCs and hPGLCs

The hESCs/human induced pluripotent stem cells (hiPSCs) on Matrigel were treated with TrypLE™ Express enzyme to enable their dissociation into single cells. The incipient mesoderm-like cells (iMeLCs) were induced by plating $2.0 \times 10^5$ to $3.0 \times 10^5$ cells per well of a human plasma fibronectin-coated 6-well plate in GK15 basal medium (GMEM with 15% KSR, 0.1 mM NEAA, one mM sodium pyruvate, two mM L-glutamine, and 0.1 mM 2-mercaptoethanol, all components purchased from Thermo Fisher

(Waltham, MA USA) containing 50 ng/ml of ACTA (R&D Systems, Minneapolis, MN, USA), three µM of CHIR99021 (TOCRIS, Bristol, UK), and 10 µM of a ROCK inhibitor (Cat.1254/10; R&D Systems, Minneapolis, MN, USA).

The hPGCLCs were induced via two methods: one was the conventional method in the ultra-low attachment surface round bottom 96-well plate (Cat. No.7007; Corning Inc., Corning, NY, USA), and the other was the new method, which added MC in the medium in the 6-well low-cell-binding plate (Cat. No.150239; Thermo Fisher, Waltham, MA, USA). With the first method, hPGCLCs were induced by planting $3.0 \times 10^3/100$ µl iMeLCs into a well of ultra-low attachment surface round bottom 96-well plate in GK15 basal medium with 500 ng/ml of BMP4 (Peprotech, Rocky Hill, NJ, USA), 100 ng/ml of LIF (Peprotech, Rocky Hill, NJ, USA), 100 ng/ml of SCF (R&D Systems, Minneapolis, MN, USA), 50 ng/ml of EGF (R&D Systems, Minneapolis, MN, USA), and 10 µM of the ROCK inhibitor (Y27632; R&D Systems, Minneapolis, MN, USA), which was hPGCLC medium. One day later, the EBs formed, and the old 50 µl medium was replaced every 2 days. With the 3D MC based induction system, two ml of the 1:7 mixture of MC and hPGCLCs medium was added into one well of 6-well low-cell-binding plate, and then, $5 \times 10^3$ to $10 \times 10^3$ total iMeLCs in one µl cell suspension were seeded in 0.35% MC medium with 50–60 replicates. One well of 6-well plate was suitable for plating $4.0 \times 10^5$ to $6.0 \times 10^5$ iMeLCs. After 1 day, the EBs formed. On the next day, the dead cells and the cell debris were eliminated after centrifugations at 600 rpm for 1 min followed by re-suspending the EBs with two ml fresh MC medium and plating into the well. Half of the medium was replaced every 2 days.

## Fluorescence-activated cell sorting

Day 4–8 EBs were washed in phosphate buffered saline (PBS) and dissociated with 0.25% trypsin/EDTA for 8–15 min at 37 °C. Dissociated cells were resuspended in fluorescence-activated cell sorting (FACS) solution consisted of 2% (v/v) fetal bovine serum in PBS. Samples were stained with APC-conjugated anti-human CD326 (EpCAM) and BV421-conjugated anti-human/mouse CD49f (INTEGRINα6) for 15 min at 4 °C. FACS analysis was performed using the FACS Calibur system (Becton Dickinson, Franklin Lakes, NJ, USA).

## Quantitative PCR

Total RNA was extracted from cells using Trizol (Life Technologies, Carlsbad, CA, USA) according to the manufacturer's instructions. Synthesis of cDNA for Q-PCR using less than two µg of purified total RNA was performed as in the technical manual using GoScript™ Reverse Transcription System (Promega, Madison, WI, USA), and the first-strand cDNAs were used for quantitative PCR analysis with $2 \times$ RealStar Green Fast Mixture (GenStar, Beijing, China) using LightCycler96™ system (Roche, Basel, Switzerland). The gene expression levels were examined by calculating ΔCt (in log2 scale) normalized to the average ΔCt values of ARBP. Error bars are mean ± SD from two or three independent experiments. The primer sequences are listed in Table S1.

## Immunofluorescence

Day 4 EBs induced from hESCs/hiPSCs were fixed in 4% paraformaldehyde in PBS for 1 h, then washed twice in PBS and incubated with 30% sucrose for 1 h at 4 °C. The samples then were embedded in OCT embedding matrix and stored at −80 °C. Subsequently, samples were sliced into seven μm cryosections by a cryostat (Leica, Heidelberger, Germany). Before immunofluorescence, slides with cryosections were air dried at room temperature for at least 15 min. Next, the slides were incubated in blocking solution (2% bovine serum albumin, 1× PBS) for 1 h at room temperature followed by incubation with primary antibodies in blocking solution overnight at 4 °C. After washing with PBS for three times, the slides were incubated with secondary antibodies in blocking solution for 1 h at room temperature, washed three times with PBS again, and then incubated with 10 μg/ml of Hoechst for 15 min at room temperature. The slides were then washed twice with PBS before being mounted in anti-fading Mounting Medium (Solarbio, Beijing, China) for confocal laser scanning microscope analysis (Carl Zeiss LSM 880, AxioObserver; Zeiss, Jena, Germany). For 5hmC staining, slides were subjected to heat-induced epitope retrieval in antigen retrieval buffers (ZSGB-BIO, Beijing, China) by a microwave oven. After retrieval, slides were cooled to room temperature and washed in PBS for three times. For H3K9me2 and H3K27me3 staining, EpCAM-/INTEGRINα6-high cells at day 4 of induction from Fy-hES-3 or YiPS were mixed with Fy-hES-3 or YiPS, respectively, at a ratio of 1:1, spread onto poly-L-lysine-coated glass slides (CITOGLAS; Citotest Labware Manufacturing Co., Jiangsu, China) by Cytospin4 (Thermo Fisher, Waltham, MA, USA) and fixed with 4% paraformaldehyde in PBS. The slides were washed with PBS for three times, incubated in blocking solution for 1 h and incubated with primary antibodies in blocking solution overnight. Primary antibodies were listed as follows: Mouse anti-Oct-3/4 (1:400, sc-5279; Santa Cruz, Dallas, TX, USA), Goat anti-Oct-3/4 (1:400, sc-5279; Santa Cruz, Dallas, TX, USA), Rabbit anti-Sox2 (1:500, Ab97959; Abcam, Cambridge, UK), Goat anti-Nanog (1:400, AF1997; R&D Systems, Minneapolis, MN, USA), Mouse anti-SSEA4 (1:500, Abcam, Cambridge, UK), Goat anti-Sox17 (1:200, AF1924; R&D Systems, Minneapolis, MN, USA), Rabbit anti-AP-2γ (1:400, sc-8977; Santa Cruz, Dallas, TX, USA), Rat anti-Blimp1 (1:200, 14-5963-82; Thermo Fisher, Waltham, MA, USA), Mouse anti-H3K9me2 (1:500, Ab1220; Abcam, Cambridge, UK), Mouse anti-H3K27me3 (1:500, Ab6002; Abcam, Cambridge, UK), Anti-5hmc (1:400, 39769; Active Motif, Carlsbad, CA, USA). Images were obtained with ZEISS LSM880 confocal microscope.

## Image analysis

Analyses and quantifications were performed with ImageJ Software (1.51k). To quantify the fluorescence intensity of H3K9me2 and H3K27me3 in the confocal images of slides spread by Cytospin4, a custom workflow was designed in the ImageJ User Guide. Briefly, each individual nucleus was selected based on its Hoechst signal. Nuclei that overlapped with TFAP2C signals were defined as hPGCLCs, while the rest were defined as hESCs/hiPSCs. The fluorescence intensities for the H3K9me2 and H3K27me3 of interest
were then measured in the two populations of nuclei, and the distributions were plotted in scatter plots. The fluorescence intensities of 5hmC of EBs sections were measured based on the above workflow as well.

## Inducing integration-free human induced pluripotent stem cells

The human testicular cells were obtained from an obstructive azoospermia (OA) patient. During two or three culture passages, cells were transfected with an episomal vector encoding six factors (Oct4, Sox2, Nanog, Lin28, Klf4, and L-MyC). After inducting in N2B27 medium for 2 weeks, the samples showed distinct small colonies were picked and expanded into PSCeasy Medium.

## Karyotyping analyses and alkaline phosphatase staining

Human induced pluripotent stem cells were incubated with culture medium containing 100 ng/ml of colchicine (Sigma-Aldrich, St. Louis, MO, USA) for 8 h. Dissociated cells were subjected to hypotonic treatment with 1% sodium citrate for 30 min at room temperature, followed by fixation in Carnoy's solution (3:1 mixture of methanol and acetic acid) and dropped onto glass slides placed on a sheet of water-soaked paper. Chromosomes were visualized by Giemsa staining. Images were captured on a Leica DM 6000 B microscope. Alkaline phosphatase (AP) staining was performed using the Leukocyte AP kit (Sigma-Aldrich, St. Louis, MO, USA).

## In vitro differentiation

For EB formation, YiPS cells were harvested by treating with collagenase IV. Cell clumps were transferred to a well of 12-well low-cell-binding plate in DMEM/F12 Medium containing 20% knockout serum replacement (KSR; Invitrogen, Carlsbad, CA, USA), two mM L-glutamine, 0.1 mM 2-mercaptoethanol (Invitrogen, Carlsbad, CA, USA), 0.1 mM NEAA, and 0.5% penicillin and streptomycin. The medium was changed every other day. After 8 days' floating culture, EBs were transferred to Matrigel-coated plates and cultured in the same medium for another 8–12 days. The RNA derived from plated EBs cells was extracted for RT-PCR.

## Teratoma formation

Human induced pluripotent stem cells were harvested by treating with collagenase IV. About $2 \times 10^6$ cells were suspended by 100 µl DF12 mixed with 50 µl Matrigel and injected subcutaneously to dorsal flank of a severe combined immunodeficiency (SCID) mouse. A total of 6–9 weeks after injection, tumors were dissected, weighed, and fixed with Carnoys fixative. Paraffin-embedded tissues were sliced and stained with hematoxylin and eosin. Sections were examined for the presence of tissue representatives of all three germ layers.

## Statistical analysis

For statistical analyses, the data of the fluorescence intensity of H3K9me2, H3K27me3, and 5hmC were analyzed by Mann–Whitney test and other data were analyzed by $t$-test.

Quantitative results were expressed as the mean ± standard deviation (SD). Statistical significance was reached when $P < 0.05$. $^*P < 0.05$, $^{**}P < 0.01$, $^{***}P < 0.001$, $^{****}P < 0.0001$.

## RESULTS

### 3D MC induction system facilitates hPGCLC differentiation

Methylcellulose is widely used in 3D culture for cell suspension and it could significantly increase the yield of target cells and generate more mature cells or tissue types (*Jiang et al., 2015*; *Kothapalli & Kamm, 2013*; *Lee & Park, 2016*; *Otsuji et al., 2014*). Therefore, we aimed to test the efficiency of a MC-based 3D hPGCLC induction system. Two male hESCs cell lines (Fy-hES-3 and Fy-hES-12, gifts from the Professor Yong Fan) were used to induce hPGCLCs in the modified system (Fig. 1A). Both cell lines shared similar cellular characteristics, and Fy-hES-3 was shown as a representative result. The hESCs were efficiently differentiated into iMeLCs, in which classical pluripotency markers including OCT4, NANOG, and SOX2 were expressed (Figs. S1A and S1B). According to the previous report (*Jiang et al., 2015*), we initially adopted 0.35% MC in PGCLC medium and found that it could effectively facilitate the iMeLCs in generating solid EBs, from which a high ratio of EpCAM-/INTEGRINα6-high cells could be obtained. In contrast, there were only adherent cells in the bottom of flat plate without MC and much lower percentage of EpCAM-/INTEGRINα6-high cells was observed (Fig. 1B; Figs. S1C and S1D). Compared to the original protocol, in which an ultra-low cell attachment U-bottom 96-well was used for the formation of EBs in hPGCLC induction (*Sasaki et al., 2015*), the hPGCLC induction efficiency in the 0.35% MC system was elevated in day 4 EBs (Fig. 1C). The effect of different concentrations of MC on inducing hPGCLCs was also tested, respectively. We found that 0.1% MC resulted in the decreased and instable differentiation efficiency compared to that of 0.35% MC, whereas the outcome of using 0.7% MC was similar to that of 0.35% MC (Figs. 1C and 1D). A little bit higher ratio of EpCAM-/INTEGRINα6-high cells could be obtained from day 4 EBs to day 8 EBs in the MC group than the U96 group (Fig. 1E). However, there was no obvious difference in the ratio of EpCAM-/INTEGRINα6-high cells at day 4 EBs via U96 method and U96 with different concentrations of MC method (Fig. S1E). In addition, when one μl cell suspensions containing $5 \times 10^3$ to $1 \times 10^4$ iMeLCs were used via 0.35% MC method, the proper size of EBs and a little bit higher ratio of hPGCLCs were obtained. In contrast, when more than $1.5 \times 10^4$ iMeLCs were seeded, EBs with bigger diameter were formed and decreased efficiency of hPGCLCs induction were observed (Fig. 1F). Notably, the increase of induction efficiency was more obvious in the day 4 but decreased in day 6 (Fig. 1G). In general, the numbers of hPGCLCs for one well of a 6-well plate via MC system were nearly twice the numbers of hPGCLCs generated from a U96 plate partially owing to more seeding cells in MC group without damaging the efficiency (Fig. S1F). Furthermore, the induction yield of hPGCLCs from the MC group was approximately 8–10 times higher than that of the U96 group at the same consumption level of hPGCLC medium (Fig. 1H; Table 1). Therefore, the 3D MC induction system combined with low cell attachment plates that had larger volume, such as 6-well plates, facilitated the generation of hPGCLCs at large scale.

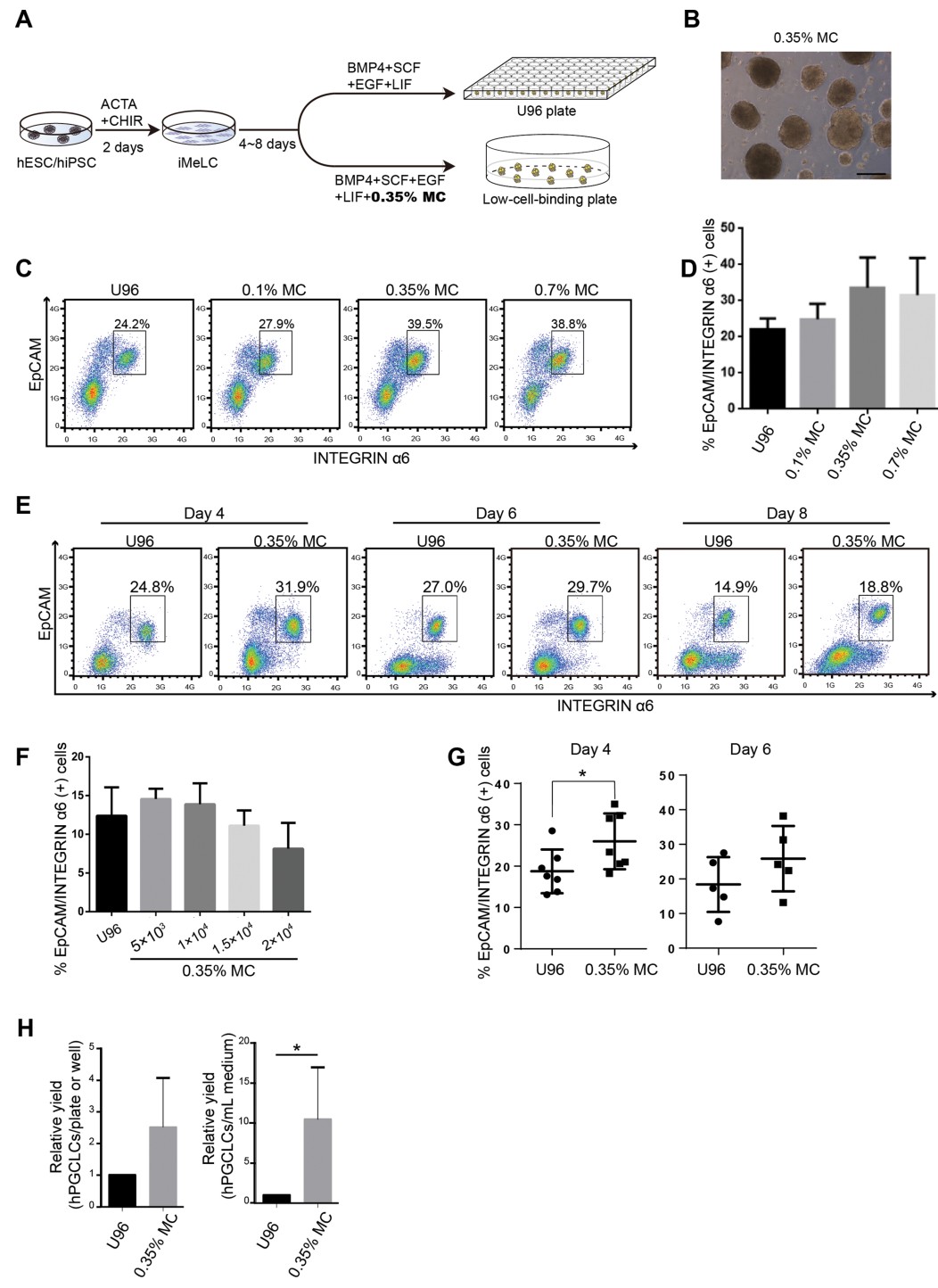

**Figure 1** **Evaluation of MC and ultra-low cell attachment U96 culture for hPGCLC differentiation.**
(A) Flow chart of hPGCLC induction system based on 0.35% MC or U96 plate. (B) Bright field image of
hPGCLCs floating aggregates cultured in hPGCLC medium containing 0.35% MC at day 4. Scale bar,
500 μm. (C) FACS analysis of the ratio of EpCAM-/INTEGRINα6-high cells of the D4 EBs of Fy-hES-3
under different MC concentrations. (D) The effect of MC concentrations on the ratio of EpCAM-/
INTEGRINα6-high cells of the D4 EBs of Fy-hES-3. (E) FACS analysis of the proportion of EpCAM-/
INTEGRINα6-high cells during the hPGCLC induction (from day 4 to day 8) of Fy-hES-3. (F) The effect
of different numbers of iMeLCs per microliter seeding suspension medium on the proportion of

**Figure 1 (continued)**
EpCAM-/INTEGRINα6-high cells of day 4 EBs from Fy-hES-12. One microliter of cell suspensions containing $5 \times 10^3$ to $2 \times 10^4$ iMeLCs were pipetted in hPGCLC medium containing 0.35% MC. (G) Differentiation efficiencies were compared between U96 method and 0.35% MC method from day 4 to day 6. $^*P < 0.05$. (H) The relative yields of hPGCLCs at day 4 via 0.35% MC method and U96 method, respectively. Left, the relative number of hPGCLCs from one U96 plate or one well of the low-attachment 6-well plate; right, the relative number of hPGCLCs for one ml medium consumption via U96 method and 0.35% MC method. Both of them were calculated at day 4 during hPGCLC induction, and the number of hPGCLCs from U96 plate was set to 1 for reference. $^*P < 0.05$. Error bars, SD ($n = 3$ measurements).

**Table 1 The yields of hPGCLCs of day 4 EBs via U96 method and 3D MC method.**

| Methods | Number of seeding cells of one U96 plate or one well of 6-well plates from MC group | Number of hPGCLCs of one U96 plate or one well of 6-well plate from MC group | Number of hPGCLCs for each experiment (3× U96 plates or 3× 6-well plates from MC group) | The ratio of hPGCLCs yield (MC group vs. U96 group for each experiment) |
|---|---|---|---|---|
| U96 | $3.5 \times 10^5$ | ~$5 \times 10^4$ | $15 \times 10^4$ | ~9.6 |
| MC | $5.0 \times 10^5$ | ~$8 \times 10^4$ | $144 \times 10^4$ | |

## The EpCAM-/INTEGRINα6-high cells from MC induction system exhibited the transcriptional characteristics of hPGCLCs

Next, we examined the gene expression profiling of the EpCAM-/INTEGRINα6-high cells from the 3D MC induction system. Q-PCR analysis showed that the gene expression profiling of day 4 EpCAM-/INTEGRINα6-high cells from the MC group were highly similar to that from the U96 group. Compared to iMeLCs, day 4 EpCAM-/INTEGRINα6-high cells showed slightly elevated expressions of *OCT4* and *NANOG*, whereas *SOX2* exhibited a greatly decreased expression pattern. With respect to the genes related to naïve pluripotency, *ZFP42* and *PRDM14* were downregulated, while *KLF4* and *ESRRB* were highly upregulated. hPGCLCs specification genes, including *SOX17, BLIMP1, NANOS3,* and *TFAP2C*, were highly elevated in EpCAM-/INTEGRINα6-high cells, whereas *STELLA* was slightly upregulated. In addition, a later PGC marker gene *DAZL* showed downregulated expression. For endoderm or mesoderm marker genes, *GATA4, T,* and *EVX1* showed obvious elevated expression in EpCAM-/INTEGRINα6-high cells. Activated in iMeLCs, *NODAL* was drastically downregulated. On the other hand, *DNMT3A* and *DNMT3B*, de novo DNA methyltransferases associated with DNA methylation in mammals, were downregulated markedly (Fig. 2A). Thus, the gene expression patterns of PGCLCs from both groups were similar, which was consistent with the previous study (*Irie et al., 2015*; *Sasaki et al., 2015*). Immunofluorescence analyses confirmed the high ratio of SOX17-positive cells in EBs and the OCT4-positive cells were also BLIMP1 and TFAP2C-positive cells (Fig. 2B). In addition, the ratio of SOX17-positive cells of day 4 EBs of 3D MC group was significantly higher than that of U96 group, which was consistent with the EpCAM/INTEGRINα6 FACS results (Figs. 1G and 2C). In summary, these findings proved that EpCAM-/INTEGRINα6-high cells from both groups shared a high similarity in terms of gene expression.

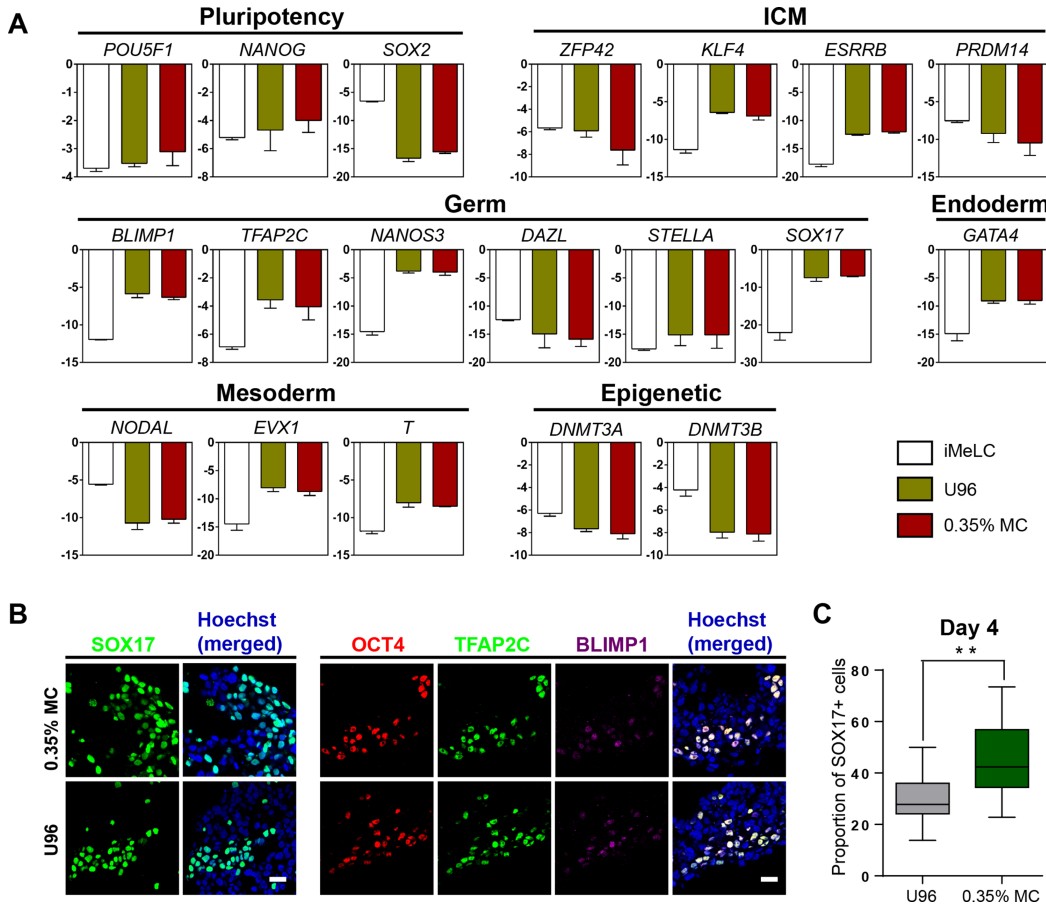

**Figure 2 Similar gene expression profile and germ cell-specific protein level of hPGCLCs generated via 0.35% MC method and U96 method.** (A) Quantitative analysis of the expression of specific genes in EpCAM-/INTEGRINα6-high cells (day 4) via U96 method and 0.35% MC method, respectively. The error bars represent mean ± SD with three biological replicates (white, iMeLC; green, U96; red, MC). (B) Immunofluorescence analysis of OCT4, TFAP2C, BLIMP1, and SOX17 on day 4 embryoids from 0.35% MC method and U96 method. The nuclei were stained with Hoechst (left). Scale bars, 20 μm. (C) The proportion of Sox17 (+) cells at day 4 EBs via 0.35% MC method and U96 method (right); gray, U96; dark green, 0.35% MC; **$P < 0.01$.

## hPGCLCs from 3D MC group exhibited key epigenetic properties and cellular characteristics similar to that from U96 group

To further confirm the identity of hPGCLCs, the epigenetic properties of hPGCLCs induced from the 3D MC group and U96 group were then evaluated. Compare to hESCs, the immunofluorescence analysis of day 4 hPGCLCs marked by TFAP2C was shown to have reduced H3K9me2 expression level and elevated H3K27me3 expression level (Fig. 3A). In addition, hPGCLCs exhibited an elevated expression level of 5hmC compare to that of hESCs (Fig. 3B). We next compared cellular viability, proliferation, apoptosis, and cell cycle of EBs between the U96 group and 3D MC group. The cellular viability of EBs could be estimated by the percentages of P1 gates (FSC/SSC) in FACS plot (Sasaki et al., 2015). FACS analysis of day 4 EBs showed that the percentages of P1 gates had no difference (73.0% vs. 75.5%) in two groups, and the ratio of P1 gates in day 6 EBs

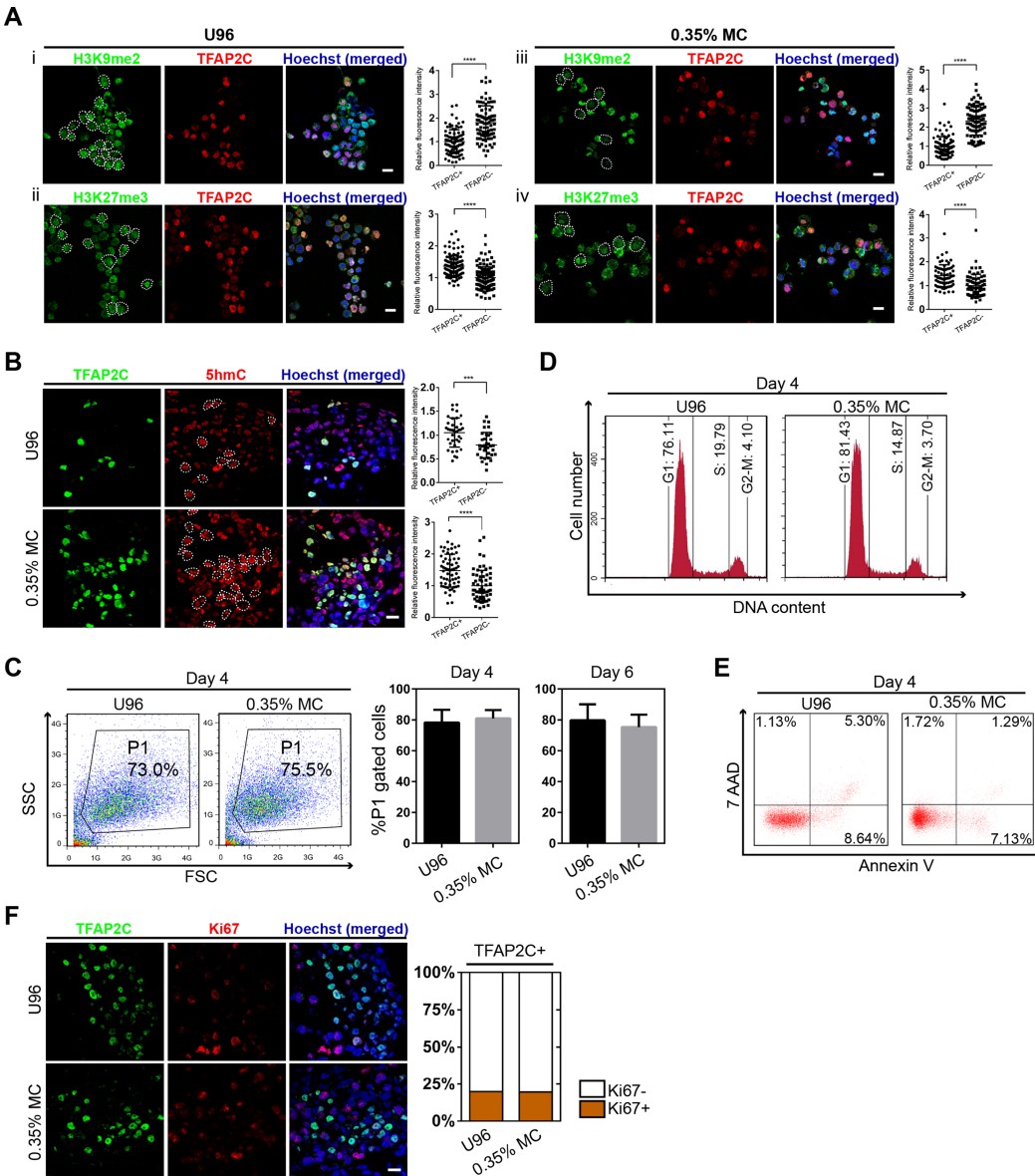

**Figure 3 Epigenetic properties and cellular states of hPGCLCs generated via 0.35% MC method and U96 method.** (A) Immunostaining of H3K9me2 and H3K27me3 of EpCAM-/INTEGRINα6-high cells from U96 (i, ii) or 0.35% MC (iii, iv) and the quantification of relative fluorescence intensity. ****$P < 0.0001$. Scale bars, 20 µm. (B) Immunostaining for 5hmC of EpCAM-/INTEGRINα6-high cells from U96 or 0.35% MC and the quantification of relative fluorescence. ***$P < 0.001$, ****$P < 0.0001$. Scale bar, 20 µm. (C) FACS analysis of cells in the P1 gates (living cells) of day 4 EBs from U96 or 0.35% MC and the statistics for ratios of P1 gates of day 4 and day 6 EBs from U96 or 0.35% MC. At least four independent experiments were performed. (D) FACS analysis of the cell cycle states of day 4 EBs from U96 and MC. (E) FACS analysis of the apoptosis of day 4 EBs from U96 and 0.35% MC using 7AAD and Annexin V. (F) Immunostaining of TFAP2C and Ki67 of day 4 EBs from U96 or 0.35% MC and the statistics for ratios of Ki67 positive or negative cells in hPGCLCs (marked by TFAP2C). Scale bar, 20 µm.

maintained the same pattern (Fig. 3C). The cell cycle also had no difference between both groups (Fig. 3D). Moreover, 7-AAD and Annexin V could be used to distinguish viable, early apoptotic, and late apoptotic/dead cells in flow cytometry (*Kim et al., 2011*).

We found that the day 4 EBs from MC group revealed a decreased ratio of apoptosis, including early- and late-stage apoptosis via FACS analysis with 7-AAD and Annexin V (Fig. 3E). Antigen Ki-67 is a nuclear protein that is associated with and may be necessary for cellular proliferation. Immunofluorescence analyses exhibited that the ki-67 positive cells accounted for similar percentages (18%) of germ cells, marked by TFAP2C, in EBs for both U96 and MC groups, suggesting that hPGCLCs in EBs under two conditions had similar proliferation capacity (Fig. 3F). Thus, EBs from the MC group showed a high quality by exhibiting a decreased percentage of apoptosis. In summary, hPGCLCs from 3D MC method showed similar epigenetic properties and cellular characteristics with that from U96 method.

## hPGCLCs derived from hiPSCs were generated in 3D MC induction system

To test the utility of the MC-based 3D hPGCLC induction, iPSCs were generated and further induced to germ cells lineage in vitro. Two hiPSC lines (named YiPS-1, YiPS-2) were induced from somatic cells of testes (donated by one patient with OA) by the ectopic expression of an episomal vector containing six factors (Oct4, Sox2, Nanog, Lin28, Klf4, and L-MyC) (Fig. 4A; Fig. S2A). hiPSC clones were positive for AP staining (Fig. S2B). PCR analysis of oriP and EBNA-1 in two hiPSC lines (YiPS-1 and YiPS-2) revealed the episomal vector was absent in the yielded hiPSCs (Fig. S2C). Since both hiPSC lines shared similar results, Yips-1 was shown as a representative for later experiments. Karyotype analysis of YiPS-1 showed 46/XY that excluded the chromosome abnormality (Fig. S2D). Q-PCR results showed YiPS cells expressed typical pluripotent marker genes, such as *OCT4*, *SOX2*, *NANOG*, *REX1*, and *DPPA3* (developmental pluripotency-associated 3) at levels similar to that of hESC H9 (Fig. S2E). Immunostaining results further confirmed that YiPS cells expressed key human pluripotent markers, including OCT4, SOX2, and SSEA4 (Fig. S2F). To determine the differentiation ability of YiPS cells in vitro, iPSCs were differentiated into EBs. YiPS cells formed EBs in suspension culture for 8 days in low-cell-binding plate (Fig. S3A). Then the EBs were transferred to Matrigel-coated plates for 8 days and attached cells on Matrigel showed epithelial morphologies (Fig. S3B) or other types of morphologies (data not shown). RT-PCR analysis showed that these differentiated cells on Matrigel expressed *AFP* (Alpha-fetoprotein, endoderm), *SOX17* (SRY-box containing gene 17, endoderm), *MSX1* (Msh homeobox 1, mesoderm), and *MAP2* (Microtubule-associated protein 2, ectoderm) (Fig. S3C). Teratomas were observed after YiPS cells were subcutaneously injected into SCID mice at week 6. Haematoxylin and eosin-stain analysis showed that the teratomas contained all three embryonic germ layers including the ectoderm, mesoderm, and endoderm (Fig. S3D).

Then, the YiPS-1 cells were induced to hPGCLCs with a 3D MC induction protocol. The iMeLCs showed a similar morphology to that of hESCs (Fy-hES-3) and expressed pluripotency markers, such as OCT4, NANOG, and SOX2 (Figs. S4A and S4B). The hiPSCs efficiently differentiated into hPGCLCs, both in the U96 group and 3D MC group (Fig. 4B). Q-PCR analysis showed that hPGCLCs derived from two groups

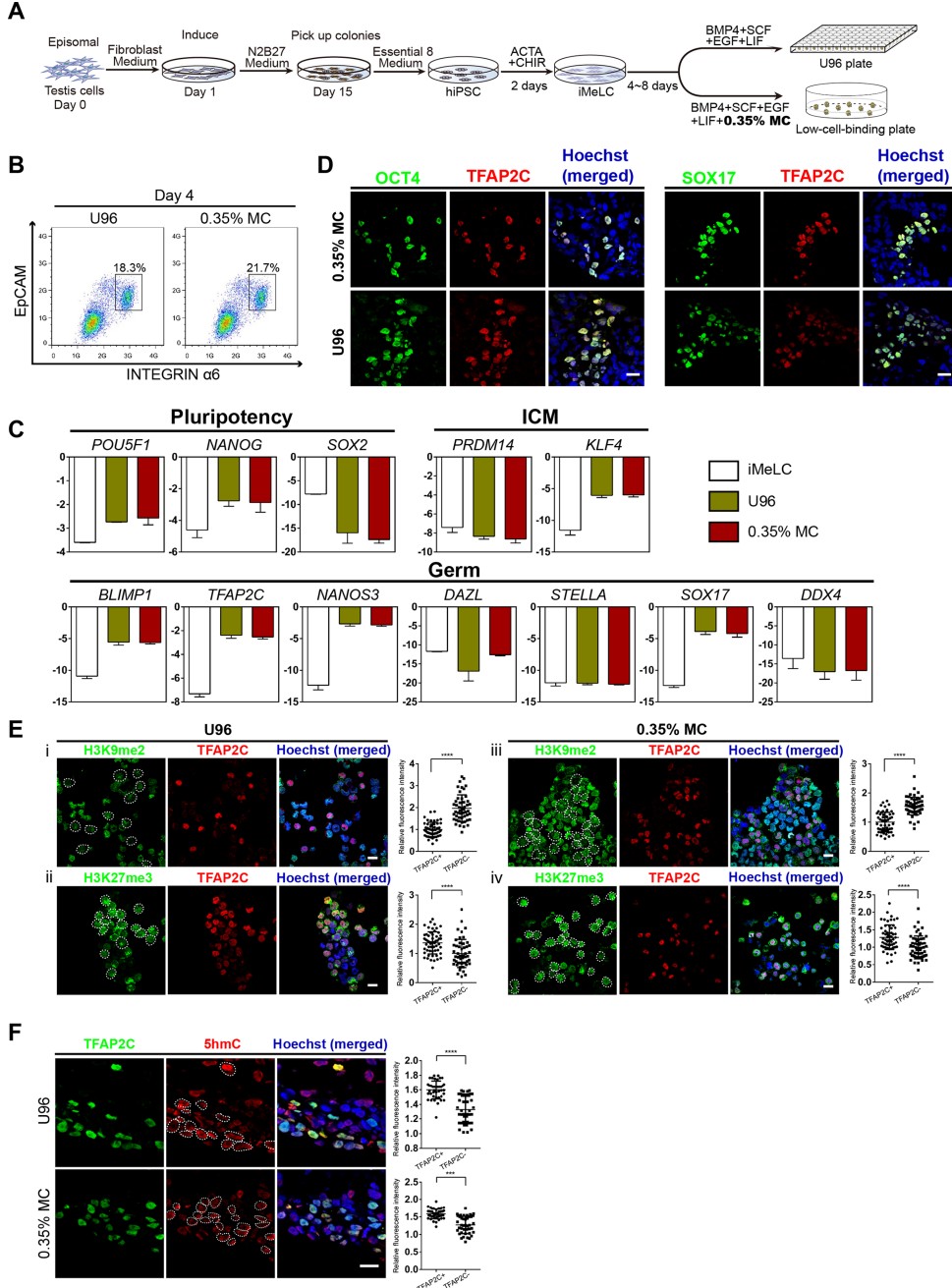

**Figure 4 High induction efficiency of hPGCLCs from hiPSCs via MC method.** (A) Flow chart of the iPSC induction from the testicular cells of one OA patient and hPGCLC differentiation from hiPSCs via U96 method or MC method, respectively. (B) FACS analysis of the expression of EpCAM and INTEGRINα6 during the hPGCLC induction (at day 4) via U96 method and 0.35% MC method, respectively. (C) Quantitative gene expression analyses of specific genes in the iMeLCs and EpCAM-/INTEGRINα6-high cells generated via U96 method and 0.35% MC method, respectively. The error bars represent mean ± SD with at least two biological replicates. (D) Immunostaining of OCT4, TFAP2C and SOX17 on day 4 EBs from 0.35% MC and U96. The nuclei were stained with Hoechst. Scale Bars, 20 μm. (E and F) Immunostaining of H3K9me2, H3K27me3 (E) and 5hmC (F) in the EpCAM-/INTEGRINα6-high cells generated via U96 method or 0.35% MC method, respectively and the quantification of relative fluorescence intensity. ***$P < 0.001$, ****$P < 0.0001$. Scale bars, 20 μm.

had a similar gene expression pattern, which was consistent with that of Fy-hES-3. Compared to iMeLCs, hPGCLCs had elevated expression levels of *OCT4*, *NANOG*, and *KLF4*, reduced expression of *PRDM14* and *SOX2*, and upregulated expression of genes specifically involved in hPGCLC specification, such as *SOX17*, *BLIMP1*, *TFAP2C*, and *NANOS3* (Fig. 4C). Immunofluorescence analyses further confirmed the expression of OCT4, TFAP2C, and SOX17 in EBs (Fig. 4D). In contrast to hiPSCs, hPGCLCs exhibited lower level of H3K9me2 and higher level of H3K27me3, which were consistent with results from hPGCLCs of Fy-hES-3 (Figs. 3A and 4E). The TFAP2C-positive cells showed a high level of 5hmC compared with that of TFAP2C negative cells (Fig. 4F). The EBs derived from YiPS-1 cells in MC group had similar cell cycle and decreased rate of apoptosis compared to EBs in U96 group (Figs. S4C and S4D). The relative induction yield of hPGCLCs from YiPS-1 cells in the MC condition was about eight times higher than that of the U96 group (Fig. S4E). In summary, about 20% EpCAM-/INTEGRINα6-high cells were induced from YiPS-1 cells via 0.35% MC method and these cells shared the characteristics of hPGCLCs including the gene expression pattern and epigenetic modification.

## DISCUSSION

The success in obtaining hPGCLCs from hPSCs provides us with a model to recapitulate the processes of hPGCLC specification and maintenance. However, unlike mPGCLCs, which can further form functional spermatozoa or germinal vesicle-stage oocytes in vivo or in vitro, hPGCLCs do not show this potential due to several challenges, such as the potential of tumour formation risk after in vivo transplantation or the limitation of materials such as somatic cells from human gonads for in vitro gamete differentiation. On the other hand, the mechanisms underlying human PGC specification were largely unknown. Although differential gene expression analysis, especially transcriptome, has provided important clues to identify the essential transcription factors during hPGCLC induction such as SOX17 and EOMES (*Irie et al., 2015*; *Kojima et al., 2017*), it is still far from figuring out the detailed mechanisms underlying germ cells specification. Further omic studies should be performed such as proteomics and ATAC-seq, which required a large number of hPGCLCs (*Gao et al., 2017*; *Guo et al., 2017*). Therefore, the efficient generation of hPGCLCs at large scale would facilitate the investigations of in vitro gamete differentiation from hPSCs and the regulatory mechanisms of hPGC/hPGCLC specification.

Although ultra-low cell attachment U-bottom 96-well plates or other similar plates have been widely used to enable the generation of EBs during the induction of hPGCLCs, this strategy restricts the seeding density of cells ($3 \times 10^5$ to $4 \times 10^5$ cells for an U96 plate) and increase the medium consumption. The process is therefore costly and labor intensive, which are not compatible with producing hPGCLCs at large scale. To generate hPGCLCs via an economic and scalable way, we employed MC to aggregate dissociated iMeLCs to produce floating EBs, in which hPGCLCs were induced effectively. Compared with the U96 method, we obtained a higher relative yields ratio of hPGCLCs via MC-based strategy gene expression levels including key germ cell markers and

epigenetic properties of hPGCLCs from the MC method are similar to hPGCLCs from the U96 system. In addition, the aggregates from the MC method showed similar cell viability states, cell cycle and proliferation capacity, and decreased rate of apoptosis compared to the aggregates from the U96 method. On the other hand, the MC method could be used to the induction of hPGCLCs from iPSCs, implying the potential clinical application value. It should be noted that our MC method depends on iMeLCs from a 2D condition, and it can be anticipated that a further elevated yield of hPGCLCs could be achieved after optimizing the culture system by performing the whole induction process from hPSCs to hPGCLCs in a 3D condition.

## CONCLUSIONS

Here, we developed a simple and efficient 3D induction system to generate hPGCLCs from hPSC at large scale. These hPGCLCs showed similar critical characteristics of those from the conventional U96 protocol, including gene expression, germ cell markers, and epigenetic properties. Furthermore, we generated iPSCs and then differentiated them into hPGCLCs efficiently, which indicated both human ESCs and iPSCs could efficiently differentiate into hPGCLCs at large scale by our new 3D system.

## ACKNOWLEDGEMENTS

We are grateful to Dr. Yong Fan for providing us with the human ESC lines Fy-hES-3 and Fy-hES-12. We thank all members of the Group for Stem Cell and Regenerative Medicine for discussion and help. We also appreciate help from Beckman Coulter Inc. for the FACS sorting.

### Funding

This work was supported by the National Basic Research Programs of China (No. 2017YFA0105001 and 2016YFC1000606), the National Natural Science Foundation Project (No. 31671544 and 31371506) and the Key Research & Development Program of Guangzhou Regenerative Medicine and Health Guangdong Laboratory (No. 018GZR110104002). The funders had no role in study design, data collection and analysis, decision to publish, or preparation of the manuscript.

### Grant Disclosures

The following grant information was disclosed by the authors:
National Basic Research Programs of China: 2017YFA0105001 and 2016YFC1000606.
National Natural Science Foundation: 31671544 and 31371506.
Key Research & Development Program of Guangzhou Regenerative Medicine and Health Guangdong Laboratory: 018GZR110104002.

### Competing Interests

The authors declare that they have no competing interests.

## Author Contributions

- Xiaoman Wang performed the experiments, analyzed the data, prepared figures and/or tables, authored or reviewed drafts of the paper, approved the final draft.
- Tingting Liao performed the experiments, contributed reagents/materials/analysis tools, authored or reviewed drafts of the paper, approved the final draft.
- Cong Wan performed the experiments, authored or reviewed drafts of the paper, approved the final draft.
- Xiaoyu Yang performed the experiments, contributed reagents/materials/analysis tools, authored or reviewed drafts of the paper, approved the final draft.
- Jiexiang Zhao performed the experiments, analyzed the data, prepared figures and/or tables, authored or reviewed drafts of the paper, approved the final draft.
- Rui Fu performed the experiments, analyzed the data, prepared figures and/or tables, authored or reviewed drafts of the paper, approved the final draft.
- Zhaokai Yao performed the experiments, analyzed the data, prepared figures and/or tables, authored or reviewed drafts of the paper, approved the final draft.
- Yaping Huang performed the experiments, authored or reviewed drafts of the paper, approved the final draft.
- Yujia Shi performed the experiments, authored or reviewed drafts of the paper, approved the final draft.
- Gang Chang analyzed the data, prepared figures and/or tables, authored or reviewed drafts of the paper, approved the final draft.
- Yi Zheng performed the experiments, authored or reviewed drafts of the paper, approved the final draft.
- Fang Luo prepared figures and/or tables, authored or reviewed drafts of the paper, approved the final draft.
- Zhaoting Liu prepared figures and/or tables, authored or reviewed drafts of the paper, approved the final draft.
- Yu Wang conceived and designed the experiments, contributed reagents/materials/analysis tools, authored or reviewed drafts of the paper, approved the final draft.
- Xinliang Mao conceived and designed the experiments, contributed reagents/materials/analysis tools, authored or reviewed drafts of the paper, approved the final draft.
- Xiao-Yang Zhao conceived and designed the experiments, contributed reagents/materials/analysis tools, authored or reviewed drafts of the paper, approved the final draft.

## Human Ethics

The following information was supplied relating to ethical approvals (i.e., approving body and any reference numbers):

All experiments were approved by the Southern Medical University ethics committee (00125817).

## Data Availability

The raw data has been supplied as a Supplementary File.

## Supplemental Information

Supplemental information for this article can be found online at http://dx.doi.org/10.7717/peerj.6143#supplemental-information.

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
