# Peer review of "Efficient generation of human primordial germ cell-like cells from pluripotent stem cells in a methylcellulose-based 3D system at large scale"

_PeerJ, doi:10.7717/peerj.6143_

## Round 0.1 · original submission · Major Revisions

Please revise the manuscript according to the reviewers' comments especially Reviewer 4.

Reviewer 1 ·

Basic reporting

The manuscript was well written.

Experimental design

The experiment was well conducted .

Validity of the findings

The manuscript reported a novel method to produce human primordial germ like cells in a large scale

Additional comments

The manuscript titled by “Efficient generation of human primordial germ cell like cells from pluripotent stem cells in a methylcellulose-based 3D system at large scale” reported a novel method to produce human primordial germ like cells in a large scale. The experiment was well conducted and the manuscript was well written. I suggest this paper should be accepted by minor revision.
Minor concern:
1. In result 1, the authors choose 0.35% MC as the optimized concerntration as refereced (Jiang et al. 2015), because 0.7% MC has a similar yield volume of PGCLC. Still I wonder about the accuracy. To make sure the experiment could be reproduced by other research team; the authors should provide more data to prove it.
2. There are no scar bars for the immunofluorescence results of Figure 2 and 3.
3. There are some grammartical problems throughout the MS, the authors should check it carefully.

·

Basic reporting

This study focuses on a new cell culture technique to induce human PGCs differentiation, it's an interesting and promising field. The format and figures in this manuscript are pretty good, and the language is also fine.

Experimental design

The project is well designed, and the content in this study is suitable to publish in PEER J.

Validity of the findings

This study provides some fundamental data of human PGCs differentiation using 3D system, which is reliable. But some of these data need further interpreted to get more precise conclusions.

Additional comments

In this manuscript Wang et al. reports a 3-D system to induce human pluripotent cells to differentiate into PGC-like cells in vitro, and compared the differentiation efficiency to traditional system. Currently, the 3-D culture system is a promising method widely used for stem cell maintenance or differentiation in vitro. However, there are still many technique obstacles in this field. The authors chose this new system to test the effect used for induction of human ESCs or iPSCs differentiation, which has significance for assisted reproductive technology. The project is well designed, and the logic is clear. Many observations are interesting, but the some of the data is not well analyzed, especially the difference of hPGCLCs derived from two systems should be discussed.
There are some suggestions for the authors:
Why the authors chose 0.1, 0.35 and 0.7% MC for research, please provide the reasons or references.
In figure 1E and F, the proportion of EpCAM/ITGA6 cells in 0.35MC group keeping on declining during culture, while the control exhibited a different characteristic. And the author chose Day 4 for further investigation, probably because the proportion of EpCAM/ITGA6 cells from two groups is significantly different in cell number. However, based on these data, we can only conclude that 0.35MC condition seams to promote differentiation faster and have a little bit higher efficiency on Day 4, but U96 condition reached the peak value of differentiation slower. The logic need be discussed here.
In figure 4C, why the hPGCLCs exhibited the decreased expression of DAZL and DDX4, while the expression of other germ line markers increased? Please analyze these data or discuss the inconsistence.
In line 329, the conclusion “All these findings demonstrated that iPSCs efficiently differentiated into hPGCLCs using the MC system” is not suitable here. The above observations only suggest that 0.35% MC system has a higher efficiency than the old system, but can not conclude that hPGCLCs were efficiently obtained, since the purity of hPGCLCs is not known.
Moreover, during differentiation in two systems, the expression of stra8 and Sycp3 (at mRNA level at least) should be detected, to make sure which stage these cell achieved.
Finally, the language is good, but more passive voice should be used in the manuscript.

Reviewer 3 ·

Basic reporting

Well.

Experimental design

Experimental design is good.

Validity of the findings

Data is well.

Additional comments

The authors developed a simple and efficient 3D induction system to generate hPGCLCs from
human pluripotent stem cells at large scale.The manuscript is well, and I suggest to accept it.

Reviewer 4 ·

Basic reporting

no comment

Experimental design

no comment

Validity of the findings

no comment

Additional comments

This manuscript developed a methylcellulose (MC)-based 3D hPGCLC induction system which can efficiently produce hPGCLCs from hESCs/hiPSCs at a large scale compared with traditional differentiation system, U bottom plate. The differentiated hPGCLCs showed similar characteristics of those from the U96 protocol, including gene expression, germ cell markers, and epigenetic properties. This differentiation system will facilitate the study of human germ cell development and stem cell-based reproductive medicine.
Specific comments:
1. This study compared the efficiencies of hPGCLCs differentiated by iMeLC in their developed system and the traditional system. However, they have three kinds of variables, i.e., different plate shapes (U bottom vs flat bottom), different seeding densities (3.5×10^5 vs 5×10^5) and different media (w/o MC vs w/ MC). It is not convincing to demonstrate that “0.35% MC” is better than “U96” by comparing these two systems with more than one experiment variable. They author should prove the effect of each factor.
2. In Fig. 1C, the author should provide critical controls including U96 plate+ different concentrations of MC, flat plate+ no MC.
3. In Fig. 1D, no technical replicates.
4. In Table 1, check the seeding # in MC group.
5. On line 251, change “30×10^4~40×10^4” to “3×10^5~4×10^5”.
6. If the abbreviations were appeared for the first time in the manuscript, it is no need to use the full names from then on. For example, embryoid bodies.

---

## Round 0.2 · accepted · Accept

All the reviewers recommend accepting your manuscript.

·

Basic reporting

The writing of this revised manuscript is good.

Experimental design

In the updated version of manuscript, several experiments were added to enhance the evidence.

Validity of the findings

Generally, the data is solid and convincible.

Additional comments

All the questions are answered, and the ambiguous points are well explained by the authors, and the updated figures in revised version also help to strengthen the conclusions. Some language errors are also revised. In all, this new version of manuscript should be accepted for publication.

Reviewer 4 ·

Basic reporting

no comment

Experimental design

no comment

Validity of the findings

no comment

Additional comments

The manuscript described a method for producing human PGCs from ESC/iPSC in bulk, which will facilitate the study on human germline development and human reproductive biology. The manuscript wrote well, and prepared solid data to prove their conclusion.